# Alien vs. Predator: Impacts of Invasive Species and Native Predators on Urban Nest Box Use by Native Birds

**DOI:** 10.3390/ani13111807

**Published:** 2023-05-30

**Authors:** Andrew M. Rogers, Françoise Lermite, Andrea S. Griffin, Berndt J. van Rensburg, Salit Kark

**Affiliations:** 1Biodiversity Research Group, School of Biological Sciences, Centre for Biodiversity and Conservation Science, University of Queensland, Brisbane, QLD 4072, Australia; 2School of Psychology, University of Newcastle, University Drive, Callaghan, NSW 2308, Australia; francoise.lermite@uon.edu.au (F.L.);; 3School of Biological Sciences, Centre for Biodiversity and Conservation Science, University of Queensland, Brisbane, QLD 4072, Australia; 4Department of Zoology, DST-NRF Centre for Invasion Biology, University of Johannesburg, Johannesburg 2006, South Africa

**Keywords:** nest boxes, urban ecology, birds, common myna, invasive species impact, predation, brushtail possums, cavity nesters, inter-specific competition, introduced species

## Abstract

**Simple Summary:**

We aimed to investigate how an invasive cavity-breeding bird, the common (Indian) myna, and a native nest predator, the common brushtail possum, influence urban nest box use by native birds in Queensland and New South Wales, Australia. We quantified nest box use by invasive and native species, assessed nesting success and failure rates, and explored what environmental factors might influence nest box occupancy and nesting success. We found that the native possums were the most common nest box users and that possum occupancy of boxes was associated with higher rates of nesting failures by all bird species. More common myna nesting attempts were observed in areas where mynas have been established longer. We found no evidence of a significant negative impact by the common myna on other birds in our study locations, which may be partly due to the low rates of use of our nest boxes by native birds. Nevertheless, better nest box design and guidelines for setting them up are needed if we aim to provide more nesting opportunities for native birds to replace the decline in big old cavity trees.

**Abstract:**

Many bird species in Australia require tree hollows for breeding. However, assessing the benefits of urban nest boxes to native birds requires frequent monitoring that allows to assess nesting success. To better understand the benefits of nest boxes for native birds, we examined the impact of local habitat characteristics, invasive species (common myna, *Acridotheres tristis*), and native mammalian predators on urban nest box use and nesting success of native birds. We installed 216 nest boxes across nine locations in southeastern Australia (S.E. Queensland and northern New South Wales) in both long-invaded sites (invaded before 1970) and more recently invaded sites (after 1990). We monitored all boxes weekly over two breeding seasons. We recorded seven bird species and three mammal species using the nest boxes. Weekly box occupancy by all species averaged 8% of all boxes, with the species most frequently recorded in the nest boxes being the common brushtail possum (*Trichosurus vulpecula*), a native cavity user and nest predator. We recorded 137 nesting attempts in the boxes across all bird species. The most frequent nesting species were the invasive alien common mynas (72 nesting attempts). We recorded an average nesting failure rate of 53.3% for all bird species. We did not record any common mynas evicting other nesting birds, and found that several native species used the same box after the common myna completed its nesting. We recorded native possums in 92% of the boxes, and possum occupancy of boxes per site was negatively correlated with bird nesting success (*p* = 0.021). These results suggest that when boxes are accessible to invasive species and native predators, they are unlikely to significantly improve nesting opportunities for native birds. To ensure efficient use of limited conservation resources, nest boxes should be designed to target species of high conservation importance and limit other species of both predators and competitors.

## 1. Introduction

Access to suitable nest sites is of critical conservation concern for cavity-breeding species, considering the loss of suitable natural tree hollows in urban environments [1,2]. While nest boxes are often used to make up for the lack of natural hollows, the use of boxes by target species is highly variable and can be influenced by environmental features and interactions with other species [3,4]. Assessing the benefits of nest boxes deployed for specific species requires frequent monitoring, especially when the intent is to improve nesting opportunities for urban birds. Nevertheless, frequent monitoring is time-consuming and costly, and boxes are often deployed without sufficient monitoring to assess their benefits to nesting birds. 

In Australia, thirty per cent of all species require tree hollows for nesting or shelter, and nest boxes are very popular as a conservation tool and among the general public. While nest box projects can be beneficial when they target specific species, the benefit of nest boxes accessible by a broad range of species is less clear [5,6,7]. Long-term studies of nest boxes that are not species-specific in native woodland have shown highly variable rates of occupancy (0–20%), with common species being the most frequent occupants [8]. Urban nest boxes in S.E. Australia were also predominantly occupied by native mammals or exotic species [9], where competition for these nesting sites can be higher than native forests due to the lack of natural nesting cavities [10,11,12]. The extent to which occupancy of nest boxes by native possums (nest predators) and invasive birds reduces nesting success for native bird species is still unclear, questioning the benefits of nest box programs for native birds in Australia. 

One of Southeast Australia’s widespread invasive cavity-nesting bird species is the common myna (*Acridotheres tristis*). The species originates from Asia and is considered one of the world’s top 100 invasive species globally [13]. In Australia, the introduced common myna occupies modified diverse climates from the tropical north-east coast, down the east coast to temperate environments and into Mediterranean climates, and reaches its highest abundance in human-dominated landscapes [14]. Despite its large invasive range [15], the impact of the common myna on native species has only been shown in a small part of its range in Australia and only on a few species [16,17]. The extent to which the common myna impacts nesting opportunities for many of the native birds the myna co-exists with across its Australian range remains a significant knowledge gap. 

In addition to invasive species, urban cavity-nesting birds must contend with native mammals, both nest predators and nest-site competitors. In native forest fragments, boxes accessible to a wide range of species are often dominated by common mammals and invasive species, reducing the number of boxes available for less common cavity nesters [18]. Native Australian mammals, such as the common brushtail possum (*Trichosurus vulpecula*) and some gliders (*Petaurus* spp.), are considered to have a significant impact in cases where they predate on threatened species [19] and where they are invasive [20,21,22]. Despite several species of possums being frequent nest box users in urban environments [23], their impacts on urban nest success have received relatively limited focus in Australia.

Adequately supplementing the loss of natural tree hollows in urban areas with nest boxes is limited by our understanding of their usage by multiple interacting urban species and how this varies across the urban matrix. Therefore, this study aimed to examine nest box use by mammals and birds in urban areas and across landscape-scale urban gradients. Specifically, we were interested in how differences in the intensity of urbanisation (urban–rural), site-level habitat structure, and the presence of nest box competitors (invasive common mynas and native mammals) affect native bird species’ use of nest boxes and their breeding success across different urban environments. 

## 2. Methods

### 2.1. Study Area and Study Design 

The study took place in the central coast region of New South Wales (N.S.W.) and S.E. Queensland (QLD; Figure 1). These regions were selected because they have comparative patterns of urbanisation and common myna invasion histories [24]. We defined ”source” sites in each region as long-invaded areas where mynas were reported before 1970 (termed “source” sites). Recently invaded “front” regions were areas in which mynas have been reportedly established (i.e., breeding) since 1990 (termed “front” sites; Figure 1). Source regions were generally located in larger cities and front regions in smaller cities/towns (Table 1), with a more rural source site (except in Gatton in Queensland). The common myna was brought to southeast QLD and Newcastle in N.S.W. via independent introduction events [24]. As a result, the birds have expanded northwards and southwards along the east coast in QLD and westward towards the inland in N.S.W. [15]. In N.S.W., we selected source sites in the greater Newcastle area, specifically in the suburbs of New Lambton and Glendale (Table 1). In addition, we selected front sites within the larger Hunter region, approximately 100 km northwest of Newcastle, in the towns of Gloucester and Krambach (Table 1). In Queensland, we selected source sites at the University of Queensland Gatton campus near Lawes and in the Brisbane suburbs of Oxley and Norman Park. Front sites were located approximately 100 km north of Brisbane in the rural towns of Dayboro and Landsborough (Figure 1). 

### 2.2. Nest Box Monitoring

Within each site, we set up 24 nest boxes across three sub-sites (eight boxes in each) selected to sample variation in urban habitat characteristics. The three sub-sites were within one km of each other and included areas with remnant vegetation, open parks, and a more urbanised area (e.g, schools, suburban streets, and caravan parks). 

The nest box dimensions were constructed of plywood and had dimensions of 400 mm (height) × 170 mm (width) × 170 mm (depth). Boxes had an entrance hole that was 65 mm in diameter. We chose these boxes as they are suitable for the common myna and several native species [16]. Nest boxes were placed at heights between three and five meters and at least 10 m from the closest box. We monitored the nesting boxes using a Signet fibre optic inspection camera mounted on a pole. We would insert the camera into the entrance hole in the nest box and take a photo or video if there was an animal in the box or if there was evidence of nesting. We obtained animal ethics for all fieldwork activities approved by the University of Queensland (Research and Integrity office, Animal Welfare Unit 454/13) and the University of Newcastle (Animal Research Authority A-2014-424).

Boxes were set up in September 2014 and monitored until June 2016. We checked boxes weekly during the breeding season of the common myna and monthly during the non-breeding season. Boxes were not checked during heavy rain to minimise disturbance to potential box occupants; heavy rain occurred on nine days in QLD and seven days in N.S.W. Monitoring was performed by a team of research students and volunteers familiar with the species likely to be in the boxes. Possums were identified to the species level in QLD but not in N.S.W. Both common brushtail possums and common ringtail possums (*Pseudocheirus peregrinus*) occur in both regions. If an animal or nest was observed in the box, that box was considered occupied or “in use” during that week. 

A nesting attempt was recorded if eggs were found in a box. We identified common myna nests by their use of nesting material (including feathers and plastic) and light blue eggs. We recorded egg colour and shape to identify other nests and watched nests until the parent birds returned or emerging adult feathers on chicks allowed species identification [25]. For each nesting attempt, after eggs hatched, the general age of chicks was recorded (i.e., not feathered, feather pins, fully feathered, and fledgling). A nesting attempt was considered a failure if eggs disappeared or chicks disappeared before they had their adult feathers. We considered a nesting attempt successful if chicks developed adult feathers and left the box [26]. Nesting success per species was calculated as the sum of successful nesting attempts over the two years. Nesting failure per species was the sum of unsuccessful nesting attempts over the two years.

### 2.3. Environmental Characteristics 

We identified vegetation diversity and structure around each nest box within a 15 m radius using circular vegetation surveys. Within each plot, we counted individual trees, measured the diameter at breast height, and identified native trees to a genus where possible. Additionally, the per cent ground cover of shrub, turf, and sealed ground (i.e., asphalt and concrete) was estimated from the centre of the plot. Finally, we counted the number of potential natural hollows in trees within each plot. Due to the difficulty in assessing whether a cavity in a tree was a suitable nesting hollow, hollow numbers at each site provided only a general measure of potential nesting site availability [27]. 

To measure variation in the remaining natural habitat area around study sites, we used remotely sensed satellite data of normalised difference vegetation index (NDVI) sensu Bino et al. (2010). We used NDVI data from NASA’s Landsat satellite, which produces images with a 30 × 30 m pixel resolution. Images for all sites were downloaded for the years 2009–2014. We used images with less than 10% cloud cover, producing 14 cloud-free images. We averaged NDVI values for pixels within a 100 m^2^ area centred on the centre of each subsite to create a single NDVI value for each subsite.

### 2.4. Statistical Analyses

For each nest box, we quantified the total number of weeks a box was occupied each year (September to August), the number of nesting attempts per nest box per breeding season, and the number of successful nesting attempts per nest box per breeding season. For birds, active nests counted as a species occupying a box, while for mammals, boxes were only considered occupied when an individual was observed in the box at the time of checking. We used generalised linear mixed models (G.L.M.s) that account for the nested study design to explore which factors influenced nest box occupancy, nesting attempts, and successful nests. The models included multiple boxes nested in different sites, locations (front and source), and regions (QLD and N.S.W.). Due to the few nesting attempts, we could not separate nesting attempts by invasive and native birds.

Response variables for the G.L.M.s included the total box occupancy, the number of nesting attempts in each box, and the total number of successful nesting attempts. We used region, location, subsite, and breeding season as random factors for each model. Explanatory variables for all models included average site level NDVI, the height of the box on the tree, the basal stem diameter of the tree (BSD), the average distance to the nearest tree, the per cent shrub ground cover in a 15 m circular plot around the tree, and the total number of natural hollows at a site (Table 1). We also included total mammal occupancy (the total number of times a mammal was recorded in a box over a breeding season) as an explanatory variable in the models for total nesting attempts, nesting success, and nesting failure. A separate model was run for each response variable with all the explanatory variables in one model. Models were fitted using the R package “glmmADMB” as it can handle data with a large number of zeros [28] in R [29]. Each model was run with a negative binomial error distribution, and all models accounted for zero inflation. Full models, including all explanatory variables, were backwards simplified by removing variables with high collinearity. Collinearity between explanatory variables was assessed with a modified version of the variance inflation factor (V.I.F.) function in the “car” package. Variables with the highest V.I.F. value were sequentially dropped from the models until the remaining variables had a V.I.F. score of less than two [30]. 

## 3. Results

We recorded seven bird species and three mammal species using our nest boxes over the two-year monitoring period, 2014–2016 (Figure 1). The two study regions shared five species in common; the common myna, rainbow lorikeet (*Trichoglossus moluccanus*), squirrel glider (*Petaurus norfolcensis*), brushtail possum, and ringtail possum (Figure 1). The white-throated treecreeper (*Cormobates leucophaea*) and pale-headed rosella were only seen in our QLD nesting boxes, while the eastern rosella (*Platycercus eximius*), crimson rosella (*Platycercus elegans*), and common (European) starling (*Sturnus vulgaris*) were only found in N.S.W. Possums occupied more boxes in QLD compared to N.S.W. (Figure 2).

Over the study period, most boxes were used at least once, including two-thirds of the nest boxes in N.S.W. (66/96; 68.75%) and over 90% of all the nest boxes in QLD (109/120; 90.8%). The average number of boxes occupied in any one week during the two breeding seasons was 17 ± 12 boxes when averaged across all sites, representing 8.0% of boxes. The maximum occupancy during any single survey was 9 boxes (9.3%) in N.S.W. and 31 nest boxes (25.8%) in QLD. 

In total, mammals were observed in our boxes 129 times and occupied 34/96 (35.4%) boxes at least once in N.S.W., and 679 times, and used 94/120 (78.3%) boxes at least once in QLD. In addition, we recorded the sequential use of boxes by mammals and birds, common myna and native birds, and consecutive native bird species. Of the 210 boxes, 46 were used consecutively by different species (Table 2). 

### 3.1. Weekly Box Occupancy

We found differences in weekly box occupancy between years in both QLD and N.S.W., with higher occupancy of mammals and more nesting attempts in the second year (Figure 2). Possums, common mynas, and native birds used the boxes differently across regions and front/source locations. In both regions, rosellas and possums were the first native species to move into boxes, while rainbow lorikeets generally did not use the boxes until the second year (Figure 2). Monitoring across both regions captured a well-defined breeding season for the common myna, which ranged from October to March (Figure 2). 

Across all nest boxes, 77 nesting attempts were recorded in N.S.W., and 60 were recorded in QLD (Figure 3). Less than half (46.7%) of all nesting attempts produced at least one fledgling. Nests were successful (i.e., fledged one chick) in 31 nesting attempts in N.S.W. (31/77 = 40.3%), while 33 nests were successful in QLD (33/60 = 55%). Rosella nests failed more often than other species (Figure 3). The common myna made more nesting attempts in N.S.W. than in QLD (40 vs. 33, respectively; Figure 2). 

### 3.2. Nesting along Urban Front–Source Sites

We found more nesting attempts across the front/source sites in the more urban, common mynas’ longer-invaded areas (92 compared to 45). This pattern held in both regions with 53 source and 23 front nesting attempts in N.S.W. and 32 source and 20 front nesting attempts in QLD. However, when only native species were counted, the study regions showed a different pattern. In N.S.W., native species nested in higher numbers in the source sites (20 sources, 15 front), while in QLD, the pattern is reversed (10 source and 20 front)

The percentage of boxes occupied by any species at least once across the entire study period was 62.5% in N.S.W. front sites and 66.6% in N.S.W. source sites. In QLD, 64.5% of all front boxes were used at least once, while 78% were used in the source sites. The total number of nesting attempts over the study period by native birds was 13 in the front sites and 21 at source sites in N.S.W. In QLD, there were 19 at the front and 9 at the source sites (Figure 2). 

We found differences in species occupying boxes in front and source sites between and within regions (Figure 2). In N.S.W., the common (European) starling only nested at front sites, the rainbow lorikeet only nested in source sites, and six species (crimson rosella, common myna, common brushtail possum, common ringtail possum, eastern rosella, and squirrel glider) were found in both locations. In QLD, the white-throated treecreeper (*Cormobates leucophaea*) was only found nesting in the front sites, while common mynas and squirrel gliders were found nesting/using boxes in the source sites, with the other four species (common brushtail possum, common ringtail possum, pale-headed rosella, and rainbow lorikeet) found across both locations. In QLD, possums of both species occupied more boxes more consistently in source sites than in front sites (Figure 2). 

The proportion of successful nesting attempts (number of nests that produced at least one fledgling/total number of nesting attempts) was similar across all front–source sites, with around 50% failing (Figure 3). However, of the seven species that nested in the boxes, common mynas had the highest number of nesting attempts and the highest total number of successful nests, with more nesting attempts in source sites than native species (Figure 3). 

### 3.3. Drivers of Box Occupancy and Nesting 

Generalised mixed models revealed a significant negative relationship between bird nesting success and total mammal occupancy of boxes at a site (z value = −2.31, *p* < 0.021; Table 3). We found no significant relationships between the measures of nest box use (total box occupancy, total nesting attempts per box, and mammal occupancy per box) and NDVI, the height of the box on the tree, the basal stem diameter of the tree (BSD), the average distance to the nearest tree, per cent shrub ground cover in a 15 m circular plot around the tree, and the total number of natural hollows at a site (Table 3). 

## 4. Discussion

The urban nest boxes in our study were primarily occupied by native mammals and introduced species, adding to the growing body of evidence that highly accessible nest boxes are unlikely to improve nesting opportunities for native birds in Australia [6,8,9,30,31]. Additionally, nest boxes such as those in this study likely facilitate native nest predators and the invasive common myna, species that are likely to be successful in urban areas even without the addition of nest boxes. The low occupancy by native birds and high variability in nest box use in our study and other work [18,32,33] highlight the complexity and significant gaps that remain when supplementing the loss of natural tree hollows in urban environments. 

Species used our nest boxes differently at the various spatial scales examined here. Variations in habitat suitability within cities [23,34] and broader landscape contexts will modify the relative availability of natural hollows, changing the importance of nest boxes for cavity-nesting species [35]. While we found no nesting attempts by common mynas in our sites along the invasion front in Queensland, we did find higher use of nest boxes by all species in the more urbanised source sites, in agreement with previous work [9]. Importantly, the variation we observed in nest box use highlights that assessing myna impacts on native species in one region or city will not necessarily apply to other areas.

We found more nesting attempts in longer-invaded areas for common myna and native species. In long-invaded areas, we observed more nesting attempts by common myna than all other birds combined. The presence of the common myna alone was not enough to deter native species from nesting, but competitive exclusion could arise when nest box occupancy reaches higher levels [36,37], as has been observed in areas where common mynas are more abundant [16]. The higher number of nesting attempts in long-invaded areas may also reflect fewer natural cavity availability given that longer-invaded areas in N.S.W. tended to be in larger cities.

We did not observe common myna displacing native birds from nest boxes, as has been observed elsewhere in the world where the common myna is invasive—such as in Israel [38,39] and in Florida, U.S.A. [40]. Our results may be due to differences in myna and native birds‘ aggression at our sites, the high numbers of empty boxes at our sites, and the low occupancy of nest boxes by native birds. Additionally, competitive interactions between common myna and native birds outside the nest boxes [12] and non-direct impacts [27] could occur, making the nest boxes less appealing for native birds. However, a commonly cited impact of the common myna, filling nest boxes with nesting material [33,40], did not deter other native parrot or rosella species from subsequently nesting in those boxes. In addition, many native species, such as the white-throated treecreeper observed in this study, fill nest cavities with material, so other native species may be adapted to cope with such behaviour. 

Weekly monitoring showed that some native species nested at low levels year-round compared to the common myna which had a seasonal breeding pattern. Therefore, any breeding-related impacts of the common myna on native species will vary depending on the length of the overlap in breeding seasons. A possible impact that warrants further study could be mynas driving a shift in the timing of nesting by native species. In N.S.W., crimson rosella nesting attempts declined as common myna nesting attempts increased over the season (Figure 2), but more data are needed to test whether this is significant. While spatial segregation in myna and native species nest sites has been observed [11,33,41], temporal shifts in breeding may also allow species to reduce or avoid competition with common mynas. Future work on competition should explore how boxes that exclude mynas influence the timing of breeding by native species. 

One of the most consistent patterns across both study regions in QLD and N.S.W. was the high number of nest failures relative to successful nesting attempts. We observed nest failures for most species, and rosellas abandoned nests at high rates. While we cannot rule out the role of other types of competition [41] and unsuitable temperatures within nest boxes [5] as additional drivers of nest failures, the high number of boxes occupied by possums is likely a factor in the high nest failure rate. Elsewhere, common brushtail possum predation of nests has a significant impact on Australia’s threatened Glossy-black cockatoo (*Calyptorhynchus lathami*) [19], and possums are important predators of bird species in New Zealand, where possums are a pest species [21,42]. On several occasions, we recorded possum predation of nests, including the predation of rainbow lorikeet chicks that were close to fledging (Figure 4). Importantly, considering interactions between native species predators in urban areas is essential when designing and deploying nest boxes [20,32], and better nest box designs are needed for urban cavity-nesting birds in Australia to reduce the impact of native predators. 

Over the two years of monitoring, we did not see a significant change in the number of boxes occupied or the number of nesting attempts. The low number of nesting attempts at each site means that the sample size was too small to test the drivers of nesting attempts for invasive and native birds separately. Several studies in Australia have reported low occupancy rates of nest boxes by target native species [5,6,18] despite a lack of natural hollows in urban areas [43]. Higher quality and better designed nest boxes are needed to improve nesting opportunities and allow better comparisons of competition. 

Our study found little evidence of competition between common myna and native species for artificial nest boxes in urban environments. The lack of direct competition in our sites may be due to the high number of nest boxes available and the low number of nesting attempts by native species. The low rates of use of our boxes by native species suggest that there was something about the nest boxes or nest box location that was unsuitable for native bird species. While parrots and other cavity-breeding birds are some of the more abundant urban species [44], the exact nesting requirements for many common species are not understood well enough to provide them with high-quality nest boxes. Despite the common myna being a globally important invasive species [13] and its large invasive range in Australia, it remains unclear what common myna impacts are on native bird breeding in our study regions. 

To better support native cavity-nesting species in urban areas, the complex interactions around nest boxes must be considered and addressed when deploying nest boxes. Reducing the impact of competition and predation through better nest box design and strategic deployment has huge conservation potential, as most of Australia’s cavity-breeding species occur within urban environments [44]. While boxes designed for specific species can be critical in supporting cavity-breeding birds [32,45], more work needs to be performed to design suitable boxes for urban cavity-nesting species in the face of invasive species and abundant mammalian nest predators in urban environments.

## Figures and Tables

**Figure 1 animals-13-01807-f001:**
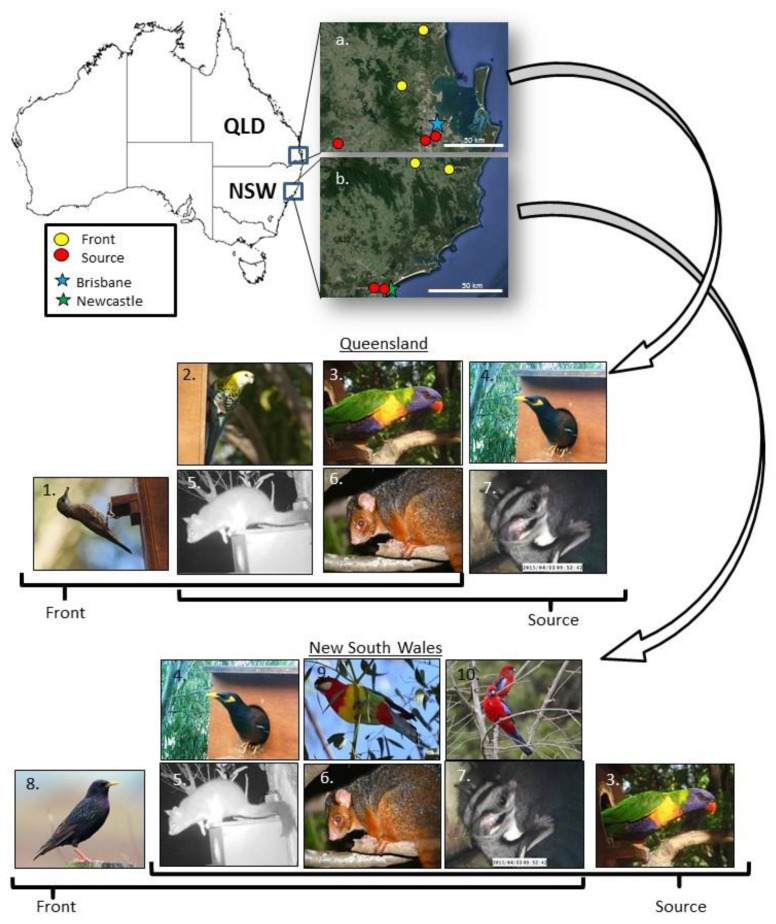
Study sites were located across two regions and across a landscape-scale invasion gradient. Study regions were (**a**) southeast Queensland (QLD) and (**b**) central New South Wales (N.S.W.). A front–source invasion gradient was identified in each region and study site (front—yellow; red—source). The species occupying the nest boxes included (1) white-throated treecreeper, (2) pale-headed rosella, (3) rainbow lorikeet, (4) common myna, (5) common brushtail possum, (6) ringtail possum, (7) squirrel glider, (8) common starling, (9) eastern rosella, and (10) crimson rosella. *Photos: A. Rogers*.

**Figure 2 animals-13-01807-f002:**
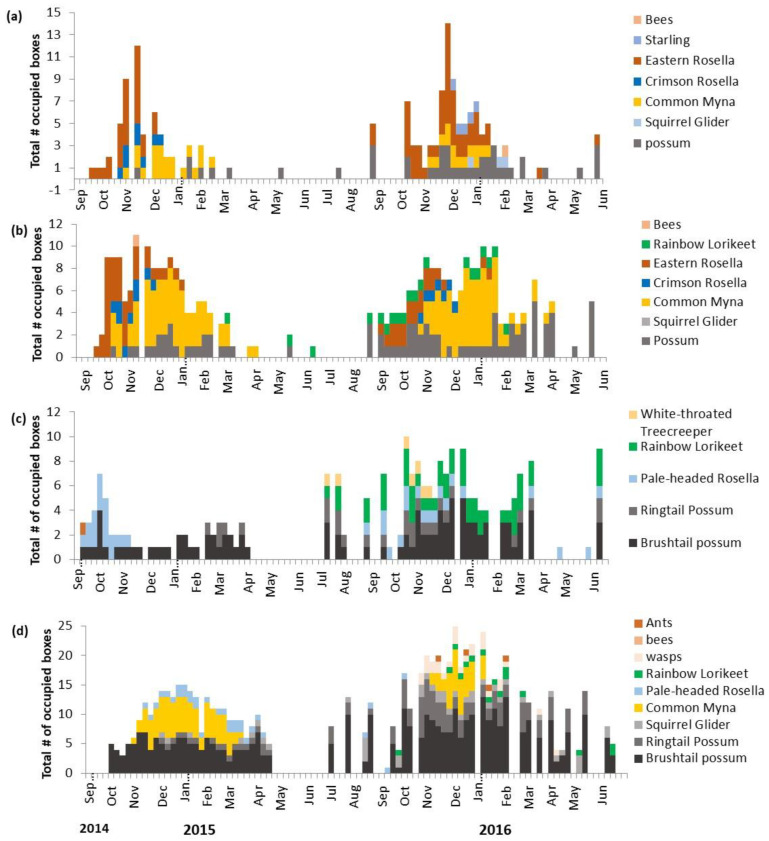
Weekly box occupancy of all 216 boxes pooled for N.S.W. (**a**) front and (**b**) source, and QLD (**c**) front and (**d**) source. Boxes were set up in late August 2014, and surveys started in September 2014. We conducted weekly surveys between September and March from 2014 to 2016, and then monitored at least once a month in winter.

**Figure 3 animals-13-01807-f003:**
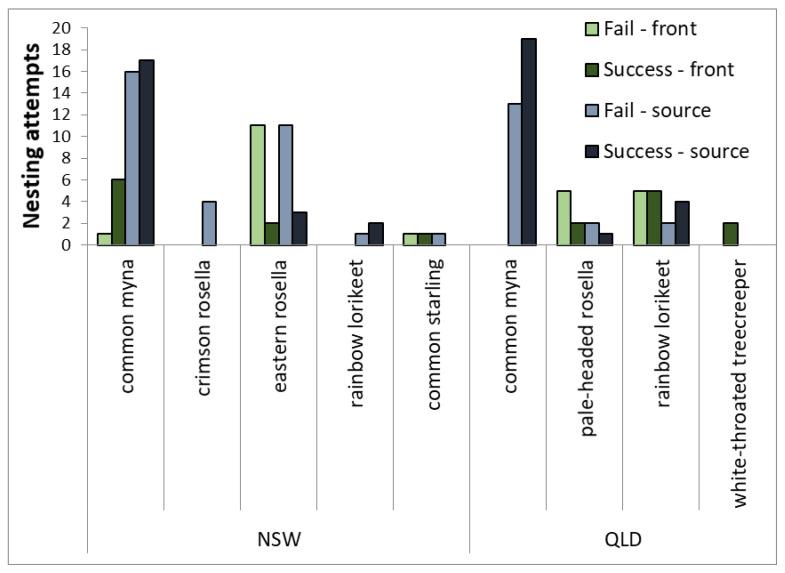
Nesting attempts and outcomes by all species across source–front locations in the two study regions. Data was pooled from both breeding seasons. Nesting attempts were considered successful if any chicks fledged from the nest. Nests were marked as a failure if eggs were abandoned, the chicks disappeared before they had adult feathers, or the chicks died.

**Figure 4 animals-13-01807-f004:**
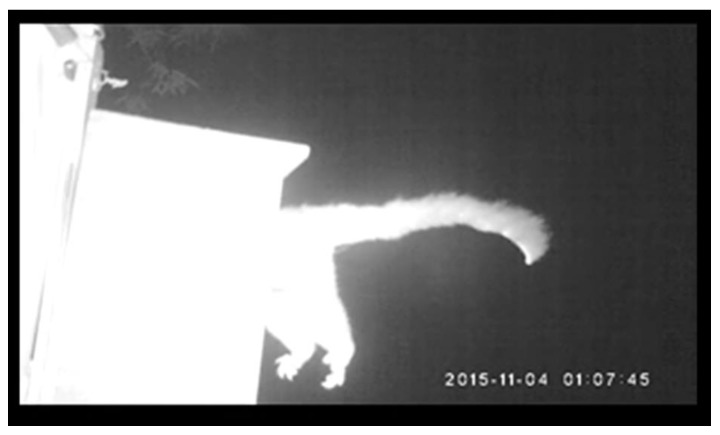
Photo of a brushtail possum (*Trichosurus vulpecula*) entering an active rainbow lorikeet nest at a Queensland front site. This photo was taken at Dayboro, at the invasion front in Queensland. The nest had three chicks before the possum was pictured entering the nest, and the chicks were discovered partially eaten after the photo was taken.

**Table 1 animals-13-01807-t001:** Site details including the region, location, human population, front/source location, NDVI for a 100 × 100 km grid around each site, the total number of natural hollows observed at a site, the per cent ground cover averaged over the eight vegetation survey plots, and the total number of trees in the vegetation survey plots.

Region	Site	Human Population (Year of the Census)	Front/Source	Sub-Environment	Average NDVI	Total Hollows	Percent Grass	Percent Sealed Surface	Percent Bush	Total Trees
N.S.W.	Gloucester	2336 (2011)	Front	Bush	0.804	0	20.6	0	79.3	334
				Park	0.698	0	56.1	20	23.8	83
				Urban	0.463	0	40	54.375	5.6	33
	Krambach	392 (2011)		Bush	0.762	0	21.8	13.75	64.3	199
				Park	0.726	0	87.5	12.5	0	91
				Urban	0.589	0	4.6	94.75	0.6	70
	Blackbutt	436,200 (2016)	Source	Bush	0.814	0	37.5	0	62.5	480
				Park	0.566	0	36.6	42.875	20.5	128
				Urban	0.311	0	27.7	70	2.25	59
	Glendale			Bush	0.591	0	62.5	0	37.5	191
				Park	0.603	0	74.7	12.625	12.6	84
				Urban	0.447	0	35.6	58.125	6.2	70
QLD	Dayboro	1692 (2011)	Front	Bush	0.611	1	25	0	75	126
				Park	0.559	1	78.7	18.125	3.1	40
				Urban	0.495	3	46.8	50	2.5	181
	Landsborough	3706 (2011)		Bush	0.719	7	38.1	9.375	52.5	136
				Park	0.5235	0	52.5	38.125	9.375	68
				Urban	0.496	5	46.2	31.875	20	102
	Norman Park	6003 (2011)	Source	Bush	0.518	0	43.7	20	36.25	76
				Park	0.554	0	90	6.875	4.375	68
				Urban	0.571	4	72.5	26.875	0.625	109
	Oxley Common	7291 (2011)		Bush	0.621	0	21.6	0.0	78.3333	168
				Park	0.616	6	55.0	2.85714	42.1429	103
				Urban	0.518	0	40.0	12.14	47.8571	302
	University of Queensland Gatton campus	304 (2011)		Bush	0.587	0	80.0	1.87	18.125	84
				Park	0.636	2	68.7	28.75	3.75	69
				Urban	0.532	7	75.0	22.5	2.5	35

**Table 2 animals-13-01807-t002:** Sequential box use by multiple species across sites. The total boxes that were occupied by multiple species. Boxes were shared at different levels across regions and source–front locations with no consistent pattern. Common mynas shared more boxes with native species than native species shared with each other, but this may be due to the higher overall occupancy of boxes by common mynas. Birds and mammals shared 26 boxes, which comprised 15% (26/166) of all the occupied boxes.

			Species Sharing Boxes		
Region	Front/Source	Site	Bird + Mammal	Myna + Native Bird	Native Birds + Native Bird
N.S.W.	front	Gloucester	3	0	0
		Krambach	0	0	0
	source	Blackbutt	5	2	1
		Glendale	3	6	2
QLD	front	Dayboro	6	0	4
		Landsborough	1	0	0
	source	Gatton	3	3	0
		Norman Park	4	0	0
		Oxley Creek Common	1	2	0
Total			26	13	7

**Table 3 animals-13-01807-t003:** Model summaries for generalised mixed models. Data were pooled for each nest box. All models used a negative binomial error distribution and accounted for zero inflation. The total number of boxes occupied by mammals at a site showed a significant negative relationship with the number of successful nesting attempts. Avg – average. * denotes a significant relationship to 0.05.

Response Variable	Explanatory Variables	Estimate	Std. Error	z-Value	Pr(>|z|)
Total box occupancy	(Intercept)	−1.997	0.397	−5.03	4.90^−07^
	Avg diameter of all trees	0.005	0.006	0.76	0.455
	Avg distance to trees	−0.043	0.037	−1.16	0.256
	Per cent shrub cover	0.000	0.006	0.04	0.971
	Total number of natural hollows	0.027	0.038	0.72	0.471
Total nesting attempts	(Intercept)	−1.924	0.367	−5.24	1.60^−07^
	Avg diameter of all trees	0.003	0.005	0.57	0.570
	Avg distance to the nearest tree	0.039	0.021	1.821	0.072
	Per cent shrub cover	−0.012	0.007	−1.741	0.082
	Total number of natural hollows	−0.106	0.062	−1.733	0.084
	Total number of boxes occupied by mammals	−0.061	0.059	−1.052	0.293
Total number of successful nesting attempts	(Intercept)	−2.191	0.437	−5.012	5.30^−07^
	Avg diameter of all trees	0.004	0.007	0.641	0.525
	Avg distance to trees	0.013	0.024	0.551	0.579
	Per cent shrub cover	−0.008	0.008	−1.07	0.284
	Total number of natural hollows	0.015	0.050	0.292	0.769
	Total number of boxes occupied by mammals	−0.659	0.285	−2.31	0.021 *
Total mammal occupancy	(Intercept)	−2.431	0.959	−2.54	0.011
	Total number of natural hollows	0.0163	0.039	0.423	0.675
	Avg diameter of all trees	0.004	0.006	0.574	0.570
	Avg distance to the nearest tree	−0.07	0.041	−1.723	0.090
	Per cent shrub cover	−0.004	0.006	−0.681	0.500

## Data Availability

Data are available at https://doi.org/10.48610/e5d3e78 (accessed on 12 May 2023).

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
