# Peer review of "Alien vs. Predator: Impacts of Invasive Species and Native Predators on Urban Nest Box Use by Native Birds"

_animals, 2023, doi:10.3390/ani13111807_

Round 1
Reviewer 1 Report
General comments (additionally to comments in pdf):
When you talk about nesting, then always birds? Maybe mention that in the methods, so it is clear for the whole manuscript.
References in text in numbers according to author guidelines?
Could you include the intensity of myna nesting into the model as explanatory variable?
Otherwise nice manuscript!

Reviewer 2 Report
The authors present an interesting study on the effect of invasive species and native predators to nest box usage (nesting attempts, nesting success). The bright side of the manuscript is to provide practical details on monitoring bird nest boxes. However, some points are missing in the manuscript. Therefore, I would like to make some suggestions to improve the quality of the paper as below:
Abstract:
“We put up 216 nest boxes nest boxes across nine locations in eastern Australia”. There are two “nest boxes”, please delete one.
Introduction:
“While nest boxes are often deployed to supplement this loss, the use of boxes by target species can be influenced by environmental features and interactions with other species” Please rephrase this sentence.
Methods:
“2.1. Study Area and study design” Please explain when was the study performed.
“2.2. Nest box monitoring”
“To identify other nests, we recorded egg colour and shape, we watched nests until the parent birds returned, or emerging adult feathers on chicks allowed identification of the species” Please add a reference (book or bird guide for identification of species, eggs, nests etc..).
“For each nesting attempt, after eggs hatched, the general age of chicks was recorded (i.e. not feathered, feather pins, fully feathered, and fledgling).” Please add references and explain how the breeding success (and/or fledgling success, hatching success, nesting success etc…) was calculated (please see: doi: 10.1080/09397140.2012.10648960).
“2.4. Statistical Analyses”
“The number of nesting attempts per box per breeding season and the number of successful nesting attempts per box per breeding season.” Please explain how the number of successful attempts/nesting success and/or breeding success was calculated in 2.2. Nest box monitoring subsection.
Results:
“We recorded seven birds and three mammal species using our nest boxes over the two-year monitoring period”. Please write the year of the study period (i.e. 2022-2023)
Please write each species' scientific names.
Discussion:
The Discussion section should be enriched with a more theoretical interpretation and relate the present results with additional concepts. For example, the discussion section can be improved with similar studies that focus on nesting success of birds and the effects of invasive species and predators from different countries. Moreover, the study results can be discussed in the framework of the impacts of invasive species on natural fauna in the broader context.
Also, the limitations of the study should be given in the discussion and/or conclusion section.
